# Contact Tracing Research: A Literature Review Based on Scientific Collaboration Network

**DOI:** 10.3390/ijerph19159311

**Published:** 2022-07-29

**Authors:** Hui Li, Yifei Zhu, Yi Niu

**Affiliations:** 1College of Information Science and Technology, Beijing University of Chemical Technology, Beijing 100029, China; 2020210440@mail.buct.edu.cn; 2China Publishing Group Digital Media Co., Ltd., Beijing 100007, China; niuyi@cnpubg.com

**Keywords:** contact tracing, scientific collaboration network, social network analysis, community detection

## Abstract

Contact tracing is a monitoring process including contact identification, listing, and follow-up, which is a key to slowing down pandemics of infectious diseases, such as COVID-19. In this study, we use the scientific collaboration network technique to explore the evolving history and scientific collaboration patterns of contact tracing. It is observed that the number of articles on the subject remained at a low level before 2020, probably because the practical significance of the contact tracing model was not widely accepted by the academic community. The COVID-19 pandemic has brought an unprecedented research boom to contact tracing, as evidenced by the explosion of the literature after 2020. Tuberculosis, HIV, and other sexually transmitted diseases were common types of diseases studied in contact tracing before 2020. In contrast, research on contact tracing regarding COVID-19 occupies a significantly large proportion after 2000. It is also found from the collaboration networks that academic teams in the field tend to conduct independent research, rather than cross-team collaboration, which is not conducive to knowledge dissemination and information flow.

## 1. Introduction

Contact tracing is an important public health tool for controlling infectious disease outbreaks [1]. In the early stage, contact tracing was used to break the transmission chains of sexually transmitted diseases (STD) [2]. In traditional contact tracing, public health officers investigate infected people to identify contacts. Contacts are then recommended to self-isolate or obtain medical evaluation and treatment [3]. The measure has successfully reduced infection transmission in many epidemics, such as severe acute respiratory syndrome (SARS) [4].

COVID-19 is a viral infectious disease caused by a virus called SARS-CoV-2. As reported by the WHO, globally, as of 17 June 2022, there have been 535,863,950 confirmed cases of COVID-19, including 6,314,972 deaths [5]. Since COVID-19 can be spread before symptoms occur or when no symptoms are present, traditional contact tracing is not good enough to deal with the pandemic [6]. Hence, health departments resort to digital tracking apps based on various technologies, such as Bluetooth, the Global Positioning System (GPS), and Wi-Fi [7].

After about seven decades of development, contact tracing today is quite different from the past both in methods and types of epidemics [8]. We made a claim based on the literature, which is considered as the primary output of corresponding research. Reasonably, when researchers make a breakthrough in some field, we know the research in advance by reading their published papers. It seems that no study has explored the overall research progress of contact tracing. Tracing chains of transmission has long been a standard part of the public health response to outbreaks, which can provide critical information to interrupt the spread of a virus [9]. Contact tracing research is not only a pandemic response but also a source of information for public health decision making. Therefore, it is of great importance to track the research status of such an interdisciplinary technique. In this paper, we use social network analysis (SNA) to analyze the network structure of the contact tracing scientific collaboration network (CTSCN) to characterize the research status of contact tracing.

The scientific collaboration network can help other researchers understand the relationships between members of the research team [10]. Understanding the structure of this network is of the first importance, as it can help us determine research priorities [11] and facilitate collaborative research programs [12].

In this study, we intend to find the answer to the following questions via CTSCN:How has the research field of contact tracing evolved? What are the characteristics of the different stages?Which disease occupies the largest proportion of contact tracing research?What is the current status of scientific research cooperation in this field?

We conducted literature research on three main scientific databases: Web of Knowledge, Scopus, and the SpringerLink database. As of 12 May 2022, 1264 related papers have been published. After identifying the types of diseases studied in the literature by subject headings, we delineated different intervals for the development of contact tracing. Next, we summarized publication volume and analyzed the co-authorship evolution over time. Finally, by constructing collaboration networks in different stages, we discussed the development characteristics of contact tracing based on the network structure and metrics.

The rest of this paper is organized as follows. Section 2 briefly reviews the development of contact tracing and the historical research of collaboration networks. Section 3 presents the collaboration network construction method and the fundamental design for contact tracing literature and explains the data and methodology of this study. Section 4 gives the results and discussion, describing the structure and metrics of CTSCN. Section 5 is the conclusion of this study.

## 2. Background

### 2.1. Contact Tracing

Contact tracing of infectious disease aims to find persons that the confirmed patient came into close contact with and give recommendations of isolating from others and monitoring symptoms. In HIV contact tracing, health department officers visit the newly infected person, making an inquiry about his/her past sexual and needle-sharing contacts [13]. Similarly, the standard practice in the application of contact tracing in tuberculosis (TB) is to assess the exposure of pulmonary TB patients to identify and treat the active or latent TB infection (LTBI) [14]. Thus far, researchers have conducted contact tracing studies during pandemics, such as H1N1 [15], Ebola [16], and MERS [17]. As for the contact tracing of COVID-19, health department officers identify all household, family, work/school, and social contacts who had come into contact with a confirmed case within the previous 14 days. A close contact for COVID-19 is defined as a person who was in close proximity (2 m or less) to a case or spent more than 15 min in an enclosed space [18].

One approach to tracing contacts for different diseases is manual (non-automatic) tracing, such as recording static personal contact information through offline and online questionnaires. Another approach is automatic tracing, such as the collection of dynamic contact events via cell phones, wearable wireless sensors, RFID, and GPS devices [19]. O’Connell et al. conducted a literature review and summarized the best practice guidelines for the design of an ideal digital contact tracing application [20]. Akinbi et al. searched the literature on contact tracing from January 2020 to January 2021, exploring challenges of the technology [21]. In addition, Hossain et al. discussed the effectiveness of contact tracing in infectious diseases to support public health decision making by reviewing the contact tracing literature in 2021 [22]. Having an overall grasp of the development trend of contact tracing, investigating diverse application scenarios and methods will help researchers to utilize past experiences to deal with unknown future pandemics.

### 2.2. Scientific Collaboration Network

Katz and Martin define scientific collaboration as a process by which researchers with a common goal work together to generate new scientific knowledge [23]. Scientific collaboration is a prerequisite for enhancing the efficiency and productivity of scientific research [24]. For example, team building through scientific analysis collaboration enables scholars with complementary skills to cooperate with each other for conducting scientific research [25]. Kretschmer researched the cooperative relationship network and proposed the concept of the co-authorship network [26]. Newman used vertices to represent scientists and edges to represent their collaborations and thus proposed the concept of the scientific collaboration network (SCN) [27]. An SCN was subsequently established among scholars in the fields of physics, biomedical research, and computer science [28].

Analyzing an SCN constructed from scientific databases helps us to track the dynamic evolution of the network [29]. Evolving over time, the structure of collaboration networks may reflect research topics, communities, and the growth/decay of the scientific field [30]. Among the currently available methods for SCN analysis, co-authorship network analysis [31], which represents authors and their co-authors as nodes and edges, is considered to be one of the most powerful methods. Since co-authorship network analysis offers a good reference for analyzing cooperation patterns [32], in this study, we will construct co-authorship networks on the basis of a literature database.

### 2.3. Social Network Analysis

Social network analysis is a powerful tool for understanding the structure and evolution of knowledge networks [33], such as exploring co-authorship networks based on digital libraries [34]. If graphs are employed to describe the relationships between social actors in social networks, it allows researchers to apply graph theory to network analysis [35]. Creating graphs is not only a kind of modeling but also an effective way to understand network behaviors and relationships between social actors [36]. For example, it facilitates calculating the average shortest path, clustering coefficient, and network efficiency [37]. Arnaboldi et al. investigated the correlation between SNA measures and scientific metrics, such as citation relationships, g-index, and h-index [38]. Moreover, as a type of network, the SCN can also be generally analyzed by calculating degree centrality, closeness centrality, betweenness centrality, and eigenvector centrality [39].

Since this research focused on exploring the dynamic evolution of the overall structure of the network, three indicators that can intuitively reflect the characteristics of the network structure were used. The first is the network density, which represents the degree of network node interaction [40]. The second is the average shortest path, representing the ability of nodes to communicate with one another [29]. The third is the clustering coefficient, which reflects the interaction probability of network nodes [41]. The above-mentioned indicators can intuitively reflect the structural characteristics of CTSCN and help us to track the dynamic evolution of the network structure.

Community detection is also an essential topic in social networks [42]. It aims to discover the community cluster from the structural information contained in the network topology, which is characterized by inner tightness and outer sparseness. Productivity within the co-authoring community is relatively high due to the close collaborative relationships [43]. Mathematically, the community detection problem is an NP-complete problem [44]. An approximate or heuristic solution to the problem is more suitable for practical applications.

Newman et al. defined the modularity Q together with a community extraction technique which finds the partition maximizing Q [45]. They also proposed a modular fastness optimization algorithm named FN [46]. Blondel et al. proposed a hierarchical agglomeration method that employs a greedy method for local optimization, combined with hierarchical clustering and a modularity optimization algorithm, thus forming hierarchical clusters [47]. To ensure that the extracted communities are connected, Traag et al. introduced the Leiden algorithm based on the Louvain algorithm [48]. Raghavan et al. applied the label propagation to community discovery and proposed a label propagation algorithm (LPA) [49]. To efficiently extract an overlapping community, Xie et al. developed a speaker–listener label propagation algorithm (SLPA) based on defining the concepts of listener and speaker [50]. Furthermore, the community detection algorithms include some other methods, such as local expansion [51], flow analysis [52], and deep learning [53].

We believe that the speed of information dissemination among all co-authors of an article is faster than that between academic groups, which is consistent with the principle of label propagation. Therefore, in this study, we adopt the community detection method combined with label propagation to extract closely connected overlapping communities.

## 3. Data and Methodology

### 3.1. Data Sources

To thoroughly understand the evolution of contact tracing research, we downloaded the data from three major scientific literature platforms: Web of Knowledge (including the Web of Science, MEDLINE, and Sci-ELO citation index scientific databases), Scopus (including specialized scientific databases, such as Elsevier, Wiley-Blackwell, and IEEE), and the SpringerLink database. We searched with the topics of “Contact Tracing”, “Contact Investigation”, and “Contact Screen” and thus obtained articles with titles, keywords, or abstracts containing these phrases. Since the study focuses on the formal research of contact tracing regarding epidemics or disease transmission, research papers and preprints were selected as the fundamental data. To avoid the literature being biased, we read the article text carefully if we could not distinguish the topic of an article from the title, keywords, and abstract. We further expunged some indirectly related literature and retained only disease-related contact tracing publications. Finally, we obtained 1264 articles as the fundamental dataset for building the collaboration networks later.

### 3.2. Methodology

In data processing, we performed identity alignment for authors. The name of an author may be inconsistent in different literature databases, e.g., “John Mickle” and “Mickle J.”. We partitioned all authors into subsets, each of which was made up of different forms of an author in the dataset. Within each subset, we compared all published articles for each author name. If some articles were identical, then we merged them as one identical author and assigned a unique identifier for these names.

Next, we conducted a subject classification. From the title, keywords, and abstract, we determined the type of disease studied for each article. When an article appeared to be an overview of contact tracing for various infectious diseases or to introduce a general model for contact tracing, we labeled it as an “infectious disease” type. We also tagged each publication by literature type (methods or studies). We assigned a unique ID to 1264 articles and generated a sequence of author identifiers corresponding to each ID.

The CTSCN is constructed with authors as nodes and mutual cooperation as edges. If two authors published an identical article, then the nodes corresponding to the two authors are connected by an edge. The weight of each edge and the multiples of co-authorship between two author nodes are positively related. The more times two authors collaborate, the greater the corresponding edge weights are.

#### 3.2.1. Network Cooperative Closeness

Network density refers to the ratio of the actual number of connected nodes to the potential maximum number of connected nodes in the network. A high network density means a high interaction between nodes, faster information dissemination, and a positive impact on network operation [40]. To measure this network density, we defined cooperative closeness (CoC) to represent the connectivity of the network. If CoC is close to 1, then any two authors in the network are closely linked to collaborate academically. Let the nodes of the network be $\vec{v}$, and the edges be $\vec{e}$. CoC can be formulated as follows: (1)$$CoC_{i}=\frac{2|\vec{e}|}{|\vec{v}\left|(\right|\vec{v}\left|-1\right)},$$
where |·| is the dimension of a vector.

#### 3.2.2. Average Shortest Path Length

The average shortest path length (ASPL) is defined as the average distance between two nodes in a network. It represents the ability of two nodes to communicate information with each other. When the paths between all nodes in the network are short, the overall information transmission efficiency of the network is high [54]. The overall CTSCN is not always a connected network. Therefore, we enumerated the ASPL of all connected subgraphs and took the average as the network ASPL. The calculation is given as follows: (2)$$ASPL=\frac{1}{n}\sum_{k=1}^{n}\sum_{i>j}^{}\frac{d(i_{k},j_{k})}{|{\vec{v}}_{k}\left|(\right|{\vec{v}}_{k}\left|-1\right)},$$
where *n* is the number of connected subgraphs of the network. ${\vec{v}}_{k}$ represents the nodes contained in the *k*-th connected subgraph, and d(·,·) calculates the distance between two nodes.

#### 3.2.3. Average Clustering Coefficient

The clustering coefficient measures network clustering and describes the symmetry of interactions among the three participants. It shows the probability that two co-authors of a scientist also co-authored an article [41]. The average clustering coefficient (ACC) is a metric defined as the average connection probability between nodes with connections in a network. Networks with a high ACC and low average path lengths are called “small-world” networks [55].
(3)$$ACC=\frac{1}{|\vec{v}|}\sum_{i=1}^{\left|v\right|}\frac{2|{\vec{w}}_{i}|}{d_{i}\left(d_{i}-1\right)},$$
where $\vec{v}$ represents the nodes, $d_{i}$ represents the degree of node *i*, and $|{\vec{w}}_{i}|$ represents the number of edges in the neighbor nodes of node *i*.

#### 3.2.4. Community Detection

Chen et al. proposed a community extraction algorithm based on SLPA optimization with label propagation only in boundary nodes [56]. We adjusted the algorithm, so the initial community was divided according to articles, as shown in Figure 1. Nodes within the initial community receive a unique label, namely the article ID. The boundary nodes represent authors with more than one article and only need to propagate labels among them. The specific steps of the CL-SLPA (SLPA with community labels) algorithm (Algorithm 1) are given as follows:
**Algorithm 1:** CL-SLPA 
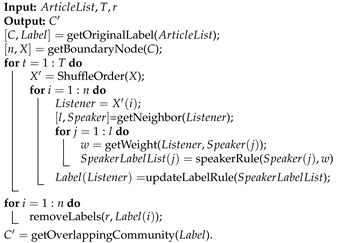


## 4. Results and Discussion

### 4.1. Literature Data Analysis

We list the year distribution of the total number of reviewed articles from 1945 to 2022 in Figure 2 and plot the corresponding literature disease type for each year. Before 1980, scholars in the field of contact tracing had only studied STDs. In the three years after 2020, a large amount of contact tracing literature on the coronavirus appeared, accounting for 92.49%, 92.51%, and 92.05% of the total literature in that year, respectively. This means that after 2020, the field of contact tracing formed a status quo that COVID-19 became the research focus.

To further study the dynamic evolution of the overall structure of the contact tracing cooperation network, we take the years 1980, 1990, 2000, 2010, and 2020 as time points. The six intervals divided by time points are 1945–1979, 1980–1989, 1990–1999, 2000–2009, 2010–2019, and 2020 to present.

#### 4.1.1. The Distribution Characteristics of Literature Data

There are only 21 articles from 1981 to 1989, and contact tracing studies of six diseases are conducted in these articles. HIV, STDs, and chlamydia are the main research diseases.

From 1990 to 1999, the number of papers grows to 81. At the same time, the types of diseases studied increases to eight. Contact tracing studies of meningitis, pelvic inflammatory disease, and CCHF appear for the first time. In addition, contact tracing for TB gradually attracts more and more attention. The proportion of the literature on tuberculosis increases from 4.76% to 23.46%, compared with that of the previous literature.

The number of contact tracing studies grows to 98 from 2000 to 2009, which contains new studies for shigellosis, SARS, and smallpox. The main research diseases of the contact tracing literature during that period are TB, chlamydia, and STDs, accounting for 28.57%, 16.33%, and 14.29% of the total literature, respectively. By 2010, contact tracing has been used for controlling many and various infectious diseases.

From 2010 to 2019, a total of 168 publications can be found for 23 diseases. The top three research subjects in this period were TB, Ebola, and HIV, accounting for 37.50%, 14.29%, and 11.31% of the total literature, respectively. This indicates that once a new epidemic emerged, scholars quickly combined it with contact tracing in response to outbreaks such as H1N1, Ebola, and MERS. After 2020, there are 872 articles on contact tracing in total. However, it is surprising that only 11 disease types are studied during this period.

Developments in the field of contact tracing initially start with STDs. They then gradually expand to HIV and chlamydia associated with STDs. In the 126 papers published from 1945 to 1999, scholars’ theoretical research and application of contact tracing are all on STD-related diseases. There are only five method papers, accounting for only 3.97% of the total number of papers. After 2000, contact tracing is used more widely for many different diseases. It can be observed that from 2000 to 2019, tuberculosis is a hot topic in the field. Specifically, there are 36 method papers among the 266 papers, accounting for 13.53%. This suggests that scholars are beginning to focus on the expansion and innovation of contact tracing methods.

After 2020, there are 133 papers regarding contact tracing methods, accounting for 15.25% of the total. It can be seen from these studies that wireless and automatic or semi-automatic contact tracing through wireless communication technology is one of the emerging and most promising technology-based solutions expected to slow the spread of COVID-19 [57]. Modern techniques lead to the emergence of a large number of contact tracing models and applications with digital tracing technology.

#### 4.1.2. Co-Author Distribution of Literature

We counted the co-author numbers of every single article. The maximum co-author number of an article is 32. Figure 3 shows the distribution of co-author numbers in the contact tracing literature over six time slots. As shown in Figure 3, between 1945 and 1999, the maximum number of co-authors is eight, but after 2000, the co-author number increases substantially. Overall, the three-people collaborative model accounts for the largest proportion, and independent research and group research with less than four people are the main co-author modes.

#### 4.1.3. Country Distribution of Literature

Figure 4 shows the country distribution of the contact tracing literature over four different periods. Between 1945 and 1980, only eight countries conduct contact tracing studies, with North America and European countries being the main study areas. From 1981 to 1999, some African countries and Australia also participate in contact tracing studies. The United Kingdom and the United States dominate the contact tracing studies, accounting for 40.2% and 19.61% of the literature. From 2000, some Asian and Latin American countries begin to conduct research on contact tracing. Thus far, the number of countries conducting research in this field has risen to 44. The countries that dominate the research from 2000 to 2019 are still the United States and the United Kingdom, accounting for 20.30% and 17.67% of the literature. Since 2020, as COVID-19 spread, 67 countries have conducted research on contact tracing to date. The top three countries in the number of documents have become the United States, the United Kingdom, and China, accounting for 20.07%, 10.55%, and 6.88% of the total.

### 4.2. Structure of CTSCN

#### 4.2.1. Basic Characteristics of the Network

Based on the authors and their collaborations, we constructed a co-authorship network for six time slots. Although Ucinet [58] and the VOS viewer are commonly used SCN visualization software applications [59], in this paper, to clearly show the network structure and evolution process of the network, we used the python package “networkx” [60] to draw the collaboration network, as shown in Figure 5. We denoted the co-authors by blue dots and the overlapping community authors by yellow stars. The labels on these nodes are the author’s identifiers. The thickness of an edge between two points depends on the degree of cooperation between two authors. If two co-authors published more articles, then the edges shown in the figure become thicker.

Table 1 lists the structural indicators of the cooperative network in each time slot. It also includes the number of overlapping communities (OCN) and the number of single-node communities (SNCN). SNCN means that the community only contains a solo author and has no connection to any other nodes.

In the two time slots of 1945–1980 and 1981–1989, most nodes in the network are isolated from others, and the network is sparse. However, the proportion of single-node communities drops from 42.11% to 33.33%, indicating that scholars begin to be more inclined to conduct collaborative research in the field of contact tracing during the period. Between 1990 and 1999, the network nodes and edges increase by more than three times compared to the previous period. At the same time, the proportion of independent research scholars gradually decreases, and the proportion of single-node communities further drops to 29.58%.

From 2000 to 2009, the number of nodes nearly doubles, and the number of edges triples, compared to the previous time slots. The co-author number has gradually increased since 2000, resulting in a rapid increase in the number of edges. The network scale of the CTSCN begins to explode in 2010, while the CoC value declines accordingly. It can be observed that the network structure is becoming more and more sparse, and the degree of cooperation in the network remains at a low level. Both academic groups and individual researchers tend to conduct independent research rather than collaboration.

In the three time slots after 2000, the proportions of single-node communities are 12.22%, 4.22%, and 10.28%, respectively. This means that the field has gradually shifted from independent research to collaborative research again. More independent researchers have been participating in COVID-19 contact tracing research. The ASPLs of these six networks are all close to 1, and the communication cost of the networks is at a low level. The three intervals after 2000 have a higher ACC and lower ASPL, with small-world characteristics.

#### 4.2.2. Network Community and Connected Subgraph Analysis

We treat the community extracted from the CTSCN as an academic team. Table 2 presents more than one disease type studied by academic teams in contact tracing. Before 1990, all teams studied only one type of disease. Between 1990 and 1999, some teams studied both STDs and HIV, while some others studied both chlamydia and STDs. In contrast to an increase in the total number of teams, only two teams working on both chlamydia and STDs can be found from 2000 to 2009, and only one team worked on both Ebola and TB from 2010 to 2019.

Since 2020, research combining COVID-19 with other diseases has become the main form of cross-disease research. As presented in Table 1, these interdisciplinary research teams account for 7.04%, 2.22%, 0.60%, and 0.49% of the total after 1990. Apparently, academic teams in the field of contact tracing tend to focus on one disease, with few crossover studies.

If two communities are in a connected subgraph of the CTSCN, it means that there is a connection between them. In other words, there is cooperation between the academic groups represented by the two communities. From the connected subgraph of the network, there is a cooperative relationship between academic teams studying various diseases, as shown in Table 3. “CSN” represents the total number of connected subgraphs across disease studies, and “Node” is the number of all nodes in the connected subgraph. From 1990 to 1999, there are five academic groups working on different diseases collaboratively. The academic teams of meningitis, pelvic inflammatory, and CCHF, which first appear during the period, do not collaborate with other teams.

A total of six new disease types are included in contact tracing studies from 2000 to 2009. Only the research teams of shigellosis and typhoid fever have collaborations with the teams of hepatitis. After 2010, only the research teams of H1N1, Ebola, Andes Virus, and COVID-19, among the 14 emerging infectious diseases, collaborate with other teams. It can be seen that only a few academic teams working on emerging epidemics have established partnerships with other experienced teams. In the four time slots after 1990, the nodes of interdisciplinary disease research account for 20.32%, 7.44%, 9.45%, and 8.47% of the total number of nodes, respectively. After 2000, only a minority of contact tracing academic groups develop partnerships across various disease types.

## 5. Conclusions

We applied social network analysis methods to the contact tracing scientific collaboration network to explore the development and the collaborative status of contact tracing research. We conclude our study as follows.

First, the number of studies related to contact tracing remained at a very low level for a long period before 2020. The peak number of articles was only 24 in 2018. This may be related to doubts about the practical significance of previous contact tracing models [61]. After 2020, the amount of literature in the field began to grow rapidly. The number of published articles in 2021 exceeded the sum total of that from 1945 to 2019. When contact tracing technology was verified in slowing the COVID-19 pandemic, the academic community began to engage with it and publish more papers.

Second, the exploration of STD tracing opened the door to research in contact tracing. Before 2010, TB, HIV, and chlamydia were the main subjects of research in contact tracing. However, research on Ebola grew drastically with a massive virus outbreak in west Africa in 2014. In the four years after 2015, contact tracing studies on the Ebola virus accounted for 66.67%, 66.67%, 50%, and 16.67% of the total literature, respectively. After 2020, contact tracing research on COVID-19 became the absolute focus in the field. From 2020 to May 2022, the proportions of research regarding the new coronavirus in each year were 92.49%, 92.51%, and 92.05%.

Third, some North American and European countries drove the initial development of contact tracing. Thus far, the number of countries participating in the study has grown from the initial 8 to 67. The United States and the United Kingdom have always dominated contact tracing research, pushing the field to a climax in the wave of the COVID-19 outbreak. China has become an important player in contributing research results in contact tracing. Although many outbreaks end naturally or can be put under control quickly, questions remain concerning how to respond scientifically to the outbreak of an unknown virus.

Fourth, by analyzing the scientific collaboration network in different time slots, we found that the scientific research cooperation network after 2000 has small-world characteristics. From the evolution of the network structure, the number of nodes was increasing, while the network density was decreasing. Academic teams in contact tracing tended to conduct independent research and weakly collaborative research. From the results of community extraction in five time slots before 2020, the proportion of single-node communities dropped from 42.11% to 4.22%, which means that the mode of independent research was gradually replaced by collaborative research. However, after 2020, the proportion of single-node communities grew to 10.28% again, probably because more and more individual scholars are involved in the research of the field.

Fifth, most research teams in the field of contact tracing studied only one disease, and a few teams studied two. Communities working on different disease types formed a densely connected subgraph, making it easier for research teams across disease areas to share knowledge. After 2000, the number of nodes contained in the connected subgraph did not exceed 10%, which does not facilitate knowledge dissemination and information flow.

In summary, contact tracing was initially carried out by studies on the control of STDs in North American and European countries. Next, more countries began studying contact tracing, and more types of infectious diseases were involved. TB, HIV, and chlamydia were the main subjects of contact tracing during this period. Subsequent pandemics, such as Ebola, continued to advance the development of contact tracing, but it was not until the outbreak of COVID-19 that research in the field culminated. The pattern that the United States and the United Kingdom were the main research countries changed recently, as China, in third place, contributed significantly to contact tracing research.

For the scale of cooperation, individual independent research was supposed to be gradually replaced by multi-person cooperation in the previous decades. However, with the outbreak of COVID-19, the field again attracted large-scale independent research scholars to join. The degree of cooperation among academic teams in this field was not high all along, corresponding to the fact that the network structure was sparse all the time.

We constructed a new analytical model of SCN, supplemented with network evaluation metrics and a modified community extraction algorithm, which helps researchers track the dynamic evolution of the network structure. The analysis of scientific research cooperation in the field of contact tracing can provide a reference for future cooperation forecasts and cooperation models in various fields different from contact tracing. To improve the effectiveness of contact tracing for the control of infectious diseases, governments should promote such interdisciplinary research. This, in return, can provide valuable information for public health decision making.

## Figures and Tables

**Figure 1 ijerph-19-09311-f001:**
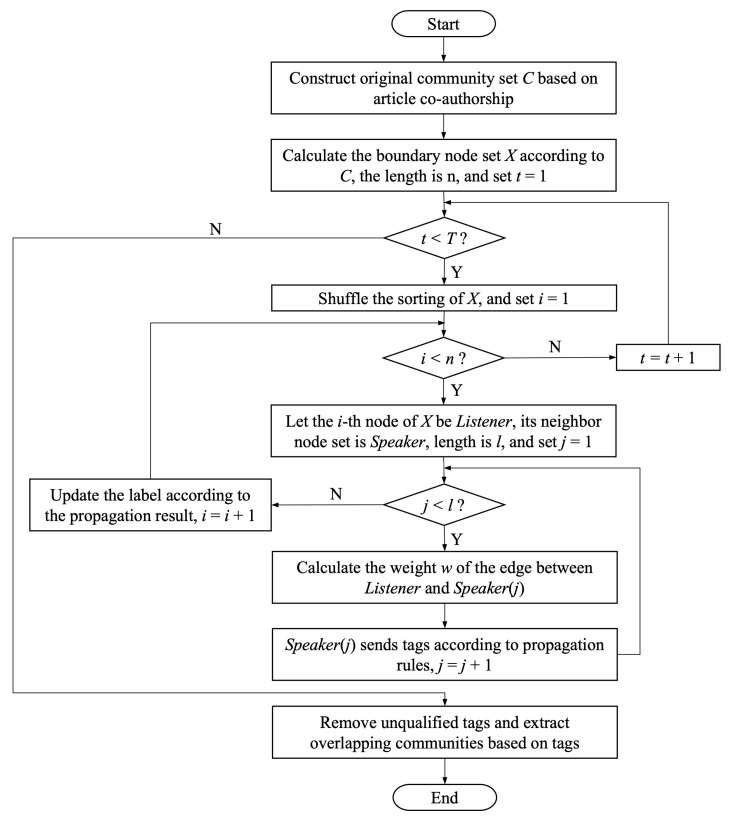
The flow chart of CL-SLPA algorithm.

**Figure 2 ijerph-19-09311-f002:**
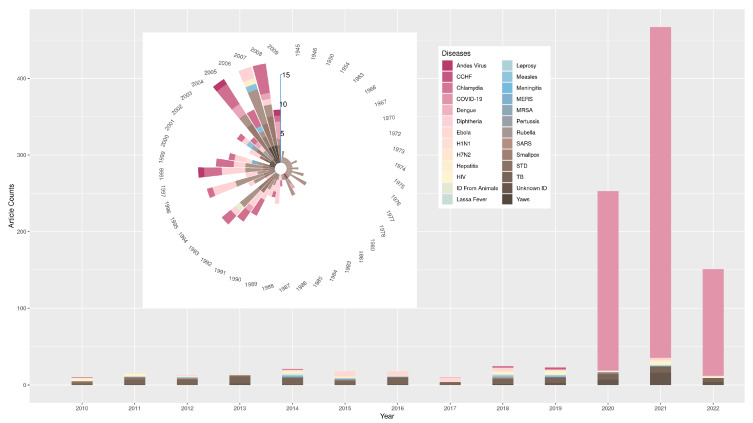
Year distribution of contact tracing literature and proportion of disease types. The circular bar plot displays annual disease types and article numbers from 1945 to 2009, while the bar graph displays annual disease types and article numbers from 2010 to 2022. In the legend, “ID” is the abbreviation of “Infective Disease”.

**Figure 3 ijerph-19-09311-f003:**
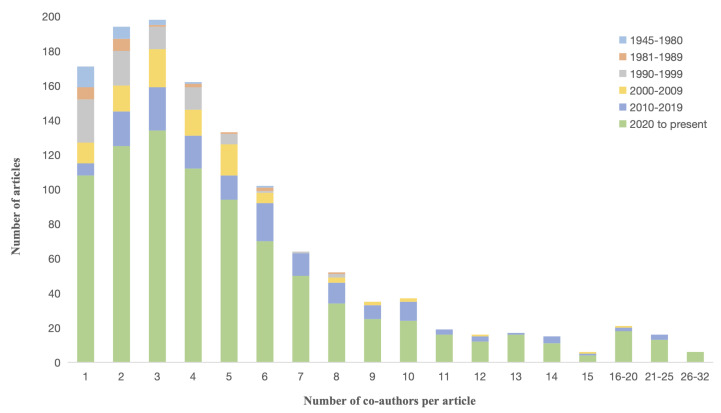
Distribution of co-author numbers in the contact tracing literature. The three-people group research is the top co-author mode.

**Figure 4 ijerph-19-09311-f004:**
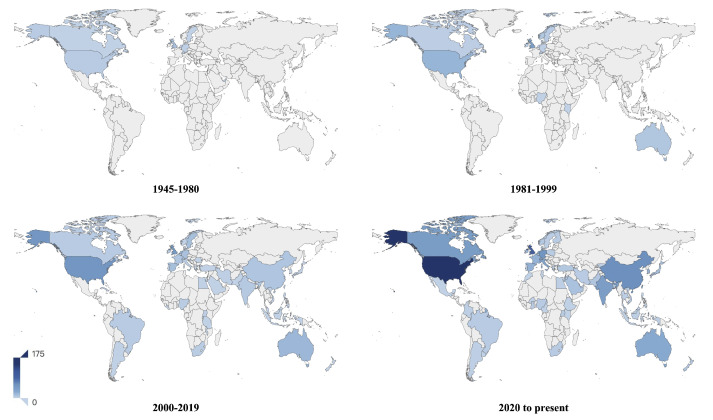
Global regional distribution map of contact tracing literature.

**Figure 5 ijerph-19-09311-f005:**
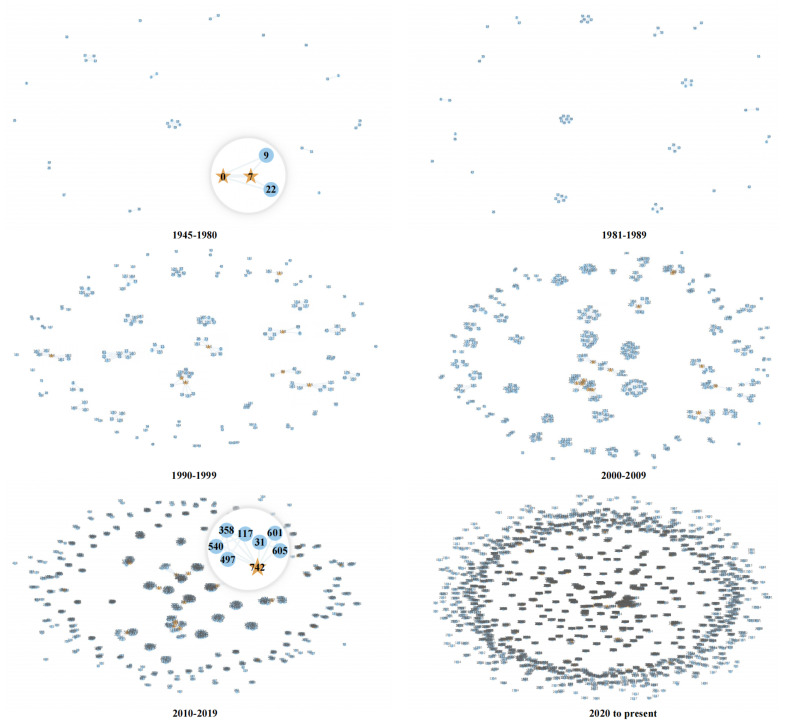
Evolution of CTSCN in six time slots.

**Table 1 ijerph-19-09311-t001:** Structural indicators of CTSCN for six year slots.

Time Slot	Article	Node	Edge	OCN	SNCN	CoC	ASPL	ACC
1945–1980	24	37	35	19	8	0.0526	1.0167	0.3833
1981–1989	21	57	90	21	7	0.0564	1.0000	0.5000
1990–1999	81	187	274	71	21	0.0158	1.0906	0.6224
2000–2009	98	363	954	90	11	0.0145	1.0526	0.8125
2010–2019	168	984	4065	166	7	0.0084	1.0481	0.8636
2020–now	872	4190	18,399	817	84	0.0021	1.0574	0.8234

**Table 2 ijerph-19-09311-t002:** Contact tracing research teams across disease types in six time slots.

Time Slot	Contact Tracing Disease Type	OCN	Node
1945–1980	-	0	0
1981–1989	-	0	0
1990–1999	(STD, HIV), (Chlamydia, STD)	5	22
2000–2009	(Chlamydia; STD)	2	8
2010–2019	(Ebola, TB)	1	3
2020–now	(COVID-19, Infectious Disease), (COVID-19, TB)	4	30

**Table 3 ijerph-19-09311-t003:** Connected subgraphs containing studies of multiple disease types in six time slots.

Time Slot	Contact Tracing Disease Type	CSN	Node
1945–1980	-	0	0
1981–1989	-	0	0
1990–1999	(STD, HIV), (Chlamydia, Hepatitis), (Chlamydia, STD, HIV), (Chlamydia, HIV), (Chlamydia, STD)	6	38
2000–2009	(Typhoid Fever, Hepatitis, Shigellosis), (Chlamydia, STD), (STD, TB)	3	27
2010–2019	(Ebola, TB), (Ebola, Andes Virus), (Meningitis, TB), (H1N1, TB), (Meningitis, H1N1), (Chlamydia, STD), (Infectious Disease, TB)	7	93
2020–now	(COVID-19, Infectious Disease, SARS), (COVID-19, Ebola), (COVID-19, Hepatitis), (COVID-19, TB, HIV), (COVID-19, HIV), (COVID-19, TB), (COVID-19, Infectious Disease)	14	355

## Data Availability

The data used to support the findings of this study were obtained from public databases.

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
