# Peer review of "Contact Tracing Research: A Literature Review Based on Scientific Collaboration Network"

_ijerph, 2022, doi:10.3390/ijerph19159311_

Round 1

Reviewer 1 Report

The COVID-19 has brought great crisis to the world. The research on contact tracing plays a positive role in tracing potential infected people in time and controlling the epidemic. This manuscript discusses the characteristics of scientific collaboration in the field of contact tracing, which is of practical significance.

This paper collects the related literature from three major scientific literature databases. Then it provides a method to describe and measure scientific cooperation network. Based on the literature, it analyzes the features of literature and illustrates the status of scientific cooperation network. The findings have enlightenment significance for reflecting the changing trend of contact tracing research and revealing the characteristics of scientific cooperation network.

It is suggested that the paper could be further improved in the following aspects:

1. Contact tracing is related to epidemiological research and public health. It is suggested to add a brief summary of the research of contact tracing in different fields in the literature review. At the same time, I suggest the authors to point out that this problem is a strong interdisciplinary problem. Therefore, it is particularly important to study the scientific cooperation in tracing close contact. That could better address the imporance of this study.

2. In the literature review, it is suggested that it’s better to summarize the measurement indicators currently which are used in previous studies of scientific cooperation network. The summary could provide more supports to this paper to illustrate the reasons that why authors select the indicators to represent the scientific cooperation network.

3. In the research design, the variables to measure the scientific cooperation network could be more comprehensive. It is suggested to include the cooperation of different countries, regions, institutions and disciplines. These findings could provide more practical contributions.

4. It is suggested to further highlight the theoretical significance and practical contributions. For example, the contribution of the proposed model. For another example, the characteristics of scientific cooperation network revealed by the findings could provide reference for cooperation prediction, cooperation mode prediction. The findings could also support government to make policies to enhance scientific cooperation in the field of contact tracing.

Author Response

Response to referee 01

We thank the reviewer for taking the time to give us these constructive comments.

[c01-01] As per the suggestion, we have added the following sentences in the introduction: “It seems that no study has explored the overall research progress of contact tracing. Tracing chains of transmission has long been a standard part of the public health response to outbreaks which can provide critical information to interrupt the spread of virus [9]. Contact tracing research is not only a pandemic response, but also a source of information for public health decision-making. Therefore, it is of great importance to track the research status of such an interdisciplinary technique.” We also revised the literature review part and added more important research to highlight the importance of our work (please see line 83-89).

[c01-02] We have added a summary of metrics used in the SCN, such as the citation relationships, g-index, and h-index. Since this research focuses on exploring the dynamic evolution of the overall structure of the network, three indicators (network density, average shortest path, and clustering coefficient) that can intuitively reflect the characteristics of the network structure are used in our work.

[c01-03] As per the suggestion, we have added a new subsection 4.1.3 and have given the country distribution of contact tracing literature over four different periods, as shown in Figure 4. “Between 1945 and 1980, only eight countries conduct contact tracing studies, with North America and European countries being the main study areas. From 1981 to 1999, some African countries and Australia also participate in contact tracing studies. The United Kingdom and the United States dominate the contact tracing studies, accounting for 40.2% and 19.61% of the literature. From 2000, some Asian and Latin American countries begin to conduct research on contact tracing. So far, the number of countries conducting research in this field has risen to 44. The countries that dominate the research from 2000 to 2019 are still the United States and the United Kingdom, accounting for 20.30% and 17.67% of the literature. After 2020, as COVID-19 spread, 67 countries have conducted research on contact tracing to date. The top 3 countries in the number of documents become the United States, the United Kingdom, and China, accounting for 20.07%, 10.55% and 6.88% of the total.” (please see subsection 4.1.3)

[c01-04] We have added the discussion in the conclusion part to emphasize the importance of our work: “We construct a new analytical model of SCN, supplemented with network evaluation metrics and a modified community extraction algorithm, which helps researchers track the dynamic evolution of the network structure. The analysis of scientific research cooperation in the field of contact tracing can provide a reference for future cooperation forecasts and cooperation models in the field. To improve the effectiveness of contact tracing for the control of infectious diseases, governments should promote such interdisciplinary research. It, in return, can provide valuable information for public health decision-making.”

Reviewer 2 Report

The authors conduct research on contact tracing which is a basic tool for slowing the spread of COVID-19 in some countries. It is an interesting review on a very important but not very deeply researched field. The authors used the scientific collaborative network technique in a review work. In my opinion, it is a good attempt. I find no obvious fault with the methods, data analysis, or conclusions. And it is also well structured. Even though, I suggest the authors make minor revisions to improve the review. My comments are given as follows:

1. In general, I think it is important to clarify the concept of scientific collaborative network.

2. The authors should provide more details about how to process the data. For example, how to deal with the situation when two or more authors have an identical name; how to post-process the search results step by step?

3. I would advise caution in making some statements, e.g. "Apparently, the network transmits information efficiently when all nodes are connected by very short paths" (Section 3.2.2). They may be true, but could be not rigorous enough out of context or in some special case.

4. Why the authors only considered the research articles?

5. I suggest the authors draw a circular barplot instead of the barplot of Fig.1.

6. "In the past three years from 2000,……" should be " from 2020 ……". Please double-check the manuscript to avoid any typos.

Author Response

Response to referee 02

We thank the reviewer for giving us these helpful comments.

[c02-01] We have clarified the concept of SCN in the sentence: “Newman used vertices to represent scientists and edges to represent their collaborations, and thus proposed the concept of scientific collaboration network (SCN).” (please see line 99-101)

[c02-02] As per the suggestion, we have added more details regarding data processing. For example, “In data processing, we perform identity alignment for authors. The name of an author may be inconsistent in different literature database, e.g. “John Mickle” and “Mickle J.”. We partition all authors into subsets, each of which is made up of different forms of an author in the dataset. Within each subset, we compare all published articles for each author name. If some articles are identical, then we merge them as one identical author and assign a unique identifier for these names.” (please see subsection 3.2)

[c02-03] We agree that it is not very rigorous. Therefore, we have removed the text "Apparently, the network transmits information efficiently when all nodes are connected by very short paths", and added couples of sentences “It represents the ability of two nodes to communicate information with each other. When the paths between all nodes in the network are short, the overall information transmission efficiency of the network is high.” (please see subsection 3.2.2)

[c02-04] Since the study focuses on the formal research of contact tracing regarding epidemics or disease transmission, the research papers and preprints are selected as the fundamental data. We have also added the explanation in subsection 3.1.

[c02-05] Thanks for this constructive comments. We have realized that a single barplot is not enough to depict the overall yearly distribution, since there are too many years. We attempt to use circular barplot instead. However, a single circular barplot is not good enough too. Finally, we parition the yearly distribution into two part, and used both barplot and circular barplot. We have redrawn the figure as Figure 2. “The circular barplot displays annual disease types and article numbers from 1945 to 2009, while the bar graph displays annual disease types and article numbers from 2010 to 2023. ” We hope that the current figure is satisfactory.

[c02-06] We have corrected the error. We have also double-checked the whole manuscript for spelling, grammar and other errors according to the suggestion.

Reviewer 3 Report

The paper is well written, characterized by a well organized structure.

The topic discussed is very interesting, and it concerns contact tracing research , that is a monitoring process that includes contact identification, listing, and follow-up, which is a key to slowing the pandemic of infectious disease such as Covid-19. A scientific network technique is used to explore the evolving history and scientific collaboration patterns of contact tracing using Social Network Analysis (SNA). Finally the authors discussed the development characteristics of contact tracing based on the network structure and metrics.     Comments:   At line 35 you affirmed : "so far, there has been no study regarding the development of contact tracing". it's not properly correct, because you can't be sure about that. I suggest to change the sentence inserting for example "it seems that no study ........"   From lines to 43 to line 47 you inserted the research questions, but they are not clear in results' section. Can you make them explicit in the results' section?   At line 162, 169 and 176 you insert  formulas. Can you describe in detail their characteristics? The descriptions are short.   At line 185 you inserted the Algorithm. It could be interesting in addition insert a Flow chart , if possible.   Figure 1 and Figure 2 describe bar plots, but they lack axis labels. Can you specify the axis' names? It could be interesting describing in details the plots in the captions.

Author Response

Response to referee 03

We thank the reviewer for taking the time to give us these helpful comments.

[c03-01] We have revised the sentence “so far, there has been no study regarding the development of contact tracing” to “It seems that no study has explored the overall evolution of contact tracing.” according to the suggestion.

[c03-02] We have answered the question asked in the introduction. We summary it here:

  • Exploration of STD tracing open the door to research in contact tracing. Before 2010, TB, HIV and chlamydia have been the main subjects of research in contact tracing. However, research on the Ebola grows drastically when a massive virus outbreak in west Africa in 2014. In the four years after 2015, contact tracing studies on Ebola virus accounted for 66.67%, 66.67%, 50% and 16.67% of the total literature, respectively. After 2020, the contact tracing research with COVID-19 becomes absolute focus in the field. From 2020 and May 2022, the proportions of research regarding the new coronavirus in each year are92.49%, 92.51% and 92.05%. (please see line 372-379)
  • By analyzing the scientific collaboration network in different time slots, we findthat the scientific research cooperation network after 2000 has the characteristics of small-world. From the evolution of network structure, the number of nodes is increasing, while the network density is decreasing. Academic teams in contact tracing tend to conduct independent research and weakly collaborative research. From the results of community extraction in five time slots before 2020, the proportion of single-node communities drops from 42.11% to 4.22%, which means that the mode of independent research is gradually replaced by collaborative research. However, after 2020, the proportion of single-node communities grows to 10.28% again, probably because that more and more individual scholars are involved in research of the field. (please see line 387-396)
  • Most research teams in the field of contact tracing studied only one disease, and a few teams studied two. Communities working on different disease types form a densely connected subgraph, making it easier for research teams across disease areas to share knowledge. After 2000, the number of nodes contained in the connected subgraph does not exceed 10%, which does not facilitate knowledge dissemination and information flow.(please see line 397-401)

Since we asked a few and found more, we did not answer the question point by point in the conclusion part. Thanks again for giving us these helpful comments.

[c03-03] As per the suggestion, we have added couples of sentences to introduce these metrics: “Network density refers to the ratio of the actual number of connected nodes to the potential maximum number of connected nodes in the network. High network density means high interaction between nodes, faster information dissemination, and a positive impact on network operation”(line 188-192); “It represents the ability of two nodes to communicate information with each other. When the paths between all nodes in the network are short, the overall information transmission efficiency of the network is high” (line 198-200); “The clustering coefficient measures network clustering and describes the symmetry of interactions among the three participants. It shows the probability that two co-authors of a scientist also co-authored an article” (line 207-209)

[c03-04] We have added a flowchart for CL-SLPA to provide more details (please see Figure 1).

[c03-05] We work hard to make these figures more descriptive. The single barplot (Figure 1 in the first version, or Figure 2 in the revised manuscript) is not enough to depict the overall yearly distribution, since there are too many years. We attempt to use circular barplot instead. However, a single circular barplot is not good enough too. Finally, we parition the yearly distribution into two part, and use both barplot and circular barplot. We have redrawn the figure as Figure 2. “The circular barplot displays annual disease types and article numbers from 1945 to 2009, while the bar graph displays annual disease types and article numbers from 2010 to 2023. ” We hope that the current figure is satisfactory. We have also redrawn the Figure 2 (in the first version, which becomes Figure 3 in the revised manuscript), adding the labels for x,y-axis, and revising the caption.

Reviewer 4 Report

The manuscript presented a very interesting study using research database on the selected topic, contact tracing. The introduction and background of the article was written well. The topic was tightly connected to the current interest of the research field. However, the reviewer requests a revision since the unclear states of methodology and discussion in the manuscript. The detail comments were listed in the following:

The reviewer has concerns on the details of methodology. It seems like the authors assigned the unique identifier to authors’ name. Was the same author has assigned more than one identifiers if the author appeared on multiple papers? If yes, would that method lead to the uncertainty in CoC? The closeness might be deviated by linking the same author (with multiple identifiers) as part of the network? In the opposite case, the name collision will play a role in the calculation. The reviewer believes this point was important. Please explain that in detail and add more discussion on this point.

The proposed method can be applied in any topic and field. In some relatively large communities, for example, chemistry, biomedical, and manufacturing, the titles and key words have strong misleading on the topic which the articles discussed in fact. The uncertainty can be introduced into the database. This was similar to one of the applied AI field challenges, the fails training data can significantly destroy the accuracy of the AI predictions. The reviewer suggests the authors to add some discussion on this point.

The study was focused on the keyword contact tracing. Is that possible in the long time past, some other terms described the similar study topic but with different terms? For example, in AI field, random forest method was called randomized decision tree back to 10 years ago. Regarding to this point, some additional discussion can be added in the manuscript.

The current discussion and conclusion were nice. The reviewer suggests the authors to add some discussion on the potential topics and directions of the future studies based on the results in the manuscript. And please briefly state the broader impact of the study in both database field and contact tracing community.

The plots in the manuscript are mostly hard to read. The reviewer understood the plots might be automatic generated by the program. However, the presentation is important. The reviewer suggests the authors to adjust the figures, such as larger fonts, adding X Y labels. Especially in Figure 3, the presentation of this figure needs to be revised. The labels on the data points are currently not readable. If the labels can be sorted to be the additional plots, please do that. Or higher dimensional figure type can be used, such as 2D/3D contours or 2D/3D scatters with color scales. And the X and Y axis would be necessary in the revised version which were absent in the current version. 

Author Response

Response to referee 04

We thank the reviewer for taking the time to give us these constructive comments.

[c04-01] As per the suggestion, we have added more details regarding identity alignment . “In data processing, we perform identity alignment for authors. The name of an author may be inconsistent in different literature database, e.g. “John Mickle” and “Mickle J.”. We partition all authors into subsets, each of which is made up of different forms of an author in the dataset. Within each subset, we compare all published articles for each author name. If some articles are identical, then we merge them as one identical author and assign a unique identifier for these names.” (please see subsection 3.2) After the treatment, the multi-name phenomenon stays in a low level. Even though, the multi-name phenomenon still exists with a small possibility for various reasons, e.g., some authors change their names intentionally. Under the circumstance, the uncertainty of CoC can be controlled to be in a very low level.

[c04-02] We have added some sentences “We search with the topic of Contact Tracing, Contact Investigation, and Contact Screen, and thus obtained articles with titles, keywords, or abstracts containing these phrases. Since the study focuses on the formal research of contact tracing regarding epidemics or disease transmission, the research papers and preprints were selected as the fundamental data. To avoid the literature be biased, we read the article text carefully if we cannot distinguish the topic of an article from title, keywords, and abstract” (please see line 160-166).

[c04-03] We have added couples of sentence in the conclusion part: “We construct a new analytical model of SCN, supplemented with network evaluation metrics and a modified community extraction algorithm, which helps researchers track the dynamic evolution of the network structure. The analysis of scientific research cooperation in the field of contact tracing can provide a reference for future cooperation forecasts and cooperation models in various field different to the contact tracing. To improve the effectiveness of contact tracing for the control of infectious diseases, governments should promote such interdisciplinary research. It, in return, can provide valuable information for public health decision-making.”

[c04-04] We work hard to make these figures more descriptive. The single barplot (Figure 1 in the first version, or Figure 2 in the revised manuscript) is not enough to depict the overall yearly distribution, since there are too many years. We attempt to use circular barplot instead. However, a single circular barplot is not good enough too. Finally, we parition the yearly distribution into two part, and used both barplot and circular barplot. We have redrawn the figure as Figure 2. “The circular barplot displays annual disease types and article numbers from 1945 to 2009, while the bar graph displays annual disease types and article numbers from 2010 to 2023. ” We hope that the current figure is satisfactory. In Figure 5 (which is Figure 3 in the first version), there are too many nodes. It is really hard to present them clearly in an A4 paper. Therefore, we revised the Figure 5 (which is Figure 3 in the first version) by adding two magnifiers in it to provide more details. Thanks again for giving us the helpful comment.

Reviewer 5 Report

This paper proposed a scientific collaborative network technique to explore the evolving history and scientific collaboration patterns of contact tracing. The main problems are as follows:

1. The method used in this paper is too simple and less innovative, which is lower than the requirements of scientific research papers.

2. I don't understand the research value and application fields of this article. Why did the author conduct this study? Can the method proposed in this paper be applied to other disciplines? Most importantly, is the result of this article helpful for contact tracing?

3. The article lacks the comparison with other existing methods.

To sum up, the innovation of the article is poor, and the research objectives and fields are not clear, so it is recommended to reject the manuscript.

Author Response

Response to referee 05

We thank the reviewer for giving us the comments.

[c05-01] Thanks for point out this. In this paper, we used social network analysis (SNA) to analyze the network structure of the contact tracing scientific collaboration network (CTSCN) to characterize the research status of the contact tracing. It is an innovative attempt in an interdisciplinary area, though we did not propose new SCN algorithms or metrics.

[c05-02] Thanks for this comments. Reasonably, when researchers make a breakthrough in some field, we know the research advance by reading their published papers. It seems that no study has explored the overall research progress of contact tracing. Tracing chains of transmission has long been a standard part of the public health response to outbreaks which can provide critical information to interrupt the spread of virus [9]. Contact tracing research is not only a pandemic response, but also a source of information for public health decision-making. Therefore, it is of great importance to track the research status of such an interdisciplinary technique. In this paper, we uses social network analysis (SNA) to analyze the network structure of the contact tracing scientific collaboration network (CTSCN) to characterize the research status of the contact tracing (please see line 34-43). We also added a paragraph in the last: “We construct a new analytical model of SCN, supplemented with network evaluation metrics and a modified community extraction algorithm, which helps researchers track the dynamic evolution of the network structure. The analysis of scientific research cooperation in the field of contact tracing can provide a reference for future cooperation forecasts and cooperation models in various field different to the contact tracing. To improve the effectiveness of contact tracing for the control of infectious diseases, governments should promote such interdisciplinary research. It, in return, can provide valuable information for public health decision-making.” (please see line 415-422)

[c05-03] Thanks for this comments. Both our work and the technique regarding the contact tracing are interdisciplinary. In addition, the main purpose of this paper is not performance improvement. Therefore, the paper lacks comparison with other methods, as the referee mentioned. Instead, we provide some comparison of metrics.

Round 2

Reviewer 4 Report

The authors addressed the comments well. The reviewer suggests an acceptance of this manuscript.

Reviewer 5 Report

Fine revision

This manuscript is a resubmission of an earlier submission. The following is a list of the peer review reports and author responses from that submission.